# A Simplified Treatment for Efficiently Modeling the Spectral Signal of Vibronic Transitions: Application to Aqueous Indole

**DOI:** 10.3390/molecules27238135

**Published:** 2022-11-22

**Authors:** Cheng Giuseppe Chen, Massimiliano Aschi, Marco D’Abramo, Andrea Amadei

**Affiliations:** 1Department of Chemistry, Sapienza University of Rome, 00185 Rome, Italy; 2Department of Chemical and Physical Sciences, University of L’Aquila, 67100 L’Aquila , Italy; 3Department of Chemical and Technological Sciences, University of Tor Vergata, 00133 Rome, Italy

**Keywords:** indole, vibronic transitions, QM/MM, electronic states, theoretical chemistry, computational chemistry, absorption spectra, emission spectra

## Abstract

In this paper, we introduce specific approximations to simplify the vibronic treatment in modeling absorption and emission spectra, allowing us to include a huge number of vibronic transitions in the calculations. Implementation of such a simplified vibronic treatment within our general approach for modelling vibronic spectra, based on molecular dynamics simulations and the perturbed matrix method, provided a quantitative reproduction of the absorption and emission spectra of aqueous indole with higher accuracy than the one obtained when using the existing vibronic treatment. Such results, showing the reliability of the approximations employed, indicate that the proposed method can be a very efficient and accurate tool for computational spectroscopy.

## 1. Introduction

Tryptophan (Trp) fluorescence has long been applied to obtain structural and dynamical information about proteins. In fact, the extensively documented sensitivity of its spectroscopic properties to environment polarity makes Trp an attractive intrinsic probe in such systems [1,2,3,4]. This environment effect can be observed in both the wavelength and the intensity of its fluorescence spectrum, and it is generally accepted that the former reflects the degree of exposure to the solvent, ranging from $\lambda_{max}$ of 308 nm in a very hydrophobic environment in azurin, to around 350 nm when completely exposed [1,4]. What still remains challenging is the possibility to interpret the spectral signal, beyond a simple qualitative assessment [5,6], providing detailed quantitative information on the molecular system of interest.

For such reasons, indole, the chromophore of the Trp amino acid, has been the subject of numerous experimental and theoretical studies [7,8,9,10,11,12,13,14,15]. Despite the relatively small size of the molecule, understanding its complex spectroscopic behavior requires accurately describing the electronic properties of its electronic ground and excited states (in particular for the excited states involving the first two gas-phase, i.e., unperturbed, spectroscopic active excited states named L${}_{b}$ and L${}_{a}$) and to properly account for the vibrational structure of the corresponding electronic transitions (i.e., the vibronic transitions). The former aspect requires the use of computationally demanding methods [12], especially in order to identify the energy minimum of the L${}_{a}$ [9,10] excited state. At the same time, it is often not feasible to characterize all the relevant vibronic transitions (possibly a huge number) by means of the existing approaches [16,17], without using proper approximations resulting in a simplified efficient treatment. Finally, it is worth remarking on the need for an explicit (atomistic) solvent model and corresponding phase-space extended sampling to accurately reconstruct the effects of the perturbation on the chromophore spectral behavior, as further suggested by a very recent paper [15]. All these issues, i.e., high-level ab-initio calculations, explicit atomistic treatment of the solvent, an extended statistical sampling of the phase space and the inclusion of all the relevant vibronic transitions, are crucial for a proper modelling of the vibronic spectra. Several efficient methods for reproducing vibronic spectra in condensed matter have appeared in the literature [17,18,19,20,21,22], as recently concisely and effectively reviewed in the work of Santoro et al. [17]. However, all such methods do not fully address all the above-mentioned important issues. In the present work, we further develop the approach presented in a previous work [23] for modelling vibronic spectra in condensed phase which was based on molecular dynamics (MD) simulations and the perturbed matrix method (PMM) [24,25,26] in combination with a widely used vibronic treatment [16]. The approach presented in this work, although maintaining the physical consistency of the theoretical–computational model, allows us to obtain rather accurate results even in the case of very complex systems, avoiding the use of highly expensive computational procedures. By employing the MD-PMM strategy and reasonable general approximations for treating the vibronic transitions, we model the environment perturbation effects on the chromophore quantum states, explicitly treating the dynamics of the atomistic water–chromophore interactions, and, hence, of the spectroscopic signal, over a much more extended statistical sampling of the system phase space (i.e., over a huge number of MD frames as obtained by a very extended MD simulation) and including a much larger number of vibronic transitions compared to the other methods. This is achieved by using, for each chromophore conformational basin (harmonic-like basin), the corresponding normal modes to provide the vibrational coordinates, assuming the quantum vibrational modes of different electronic eigenstates with similar frequencies to be identical, and, therefore, replacing the highly expensive integral calculations required to evaluate the vibrational eigenstate overlaps with the computation of undemanding single-mode integrals. In fact, similarly to the recently introduced approach of Santoro et al. [17], our approach allows us to distinguish between classical-like and quantum coordinate subspaces, permitting the use of only the quantum-mode transitions and, hence, of the minimum energy structures within the quantum internal coordinate subspace only, regardless of the positions of the semiclassical internal coordinates (e.g., at non-stationary points). However, in our approach, the use of the local full-space normal modes providing, within rather general and typically accurate approximations (see Appendix B), a proper definition of the quantum subspace (avoiding any a-priori possibly arbitrary choice) makes it possible to very easily separate and uncouple the corresponding vibronic transitions. Interestingly, such a feature of the method allows the use of the electronic state properly minimized reference configuration (e.g., by means of EOM-CCSD, necessary to obtain fully reliable results) still calculating the corresponding harmonic vibrational modes via a less accurate but computationally much more affordable method (e.g., TD-DFT). Note that the use of a lower level of theory for calculating the Hessian at the obtained higher level optimized geometry typically results in a few negative Hessian eigenvalues (i.e., the higher level optimized geometry is a saddle point for the lower level calculations), corresponding to a few classical-like modes (low-frequency modes). The efficiency of the proposed approach, which allows us to adopt expensive ab-initio calculations (in this case EOM-CCSD/6-311+G(d)) for obtaining the electronic properties and the minimized structures for each electronic state, accurately describe the perturbation effects over a very extended MD sampling and treat a huge number of vibronic transitions, made us confident about using this method to overcome the shortcomings that current strategies may incur (i.e., inaccurate perturbation treatment, insufficient phase space sampling, insufficient inclusion of vibronic transitions and inadequate description of the excited states [12]). The reported results show that the proposed extension of our MD-PMM approach, based on the use of the simplified vibronic treatment, can provide accurate predictions of both the absorption and fluorescence spectra of indole in aqueous solution, with a marked improvement compared to the results obtained using the previously employed vibronic treatment.

## 2. Theory

### 2.1. The Perturbed Vibronic Eigenstate

As usual for mixed quantum–classical models, we subdivide the system into the quantum center (QC), the subpart to be treated at quantum level, and the purely classical perturbing environment. The accurate evaluation of the QC electronic Hamiltonian operator (including the QC–environment interaction) and, hence, of the (perturbed) electronic eigenstates (electronic states) and properties can be used, in principle, to obtain any QC quantum process, possibly involving both the electronic and nuclear quantum degrees of freedom. It is worth noting that the QC nuclear semiclassical degrees of freedom, beyond the roto-translational coordinates, possibly include both the conformational coordinates corresponding to highly anharmonic degrees of freedom and providing large structural changes, and the semiclassical vibrational coordinates responsible, at each conformational configuration, for relatively small harmonic or quasi-harmonic structural fluctuations. Within the PMM framework, the electronic eigenstates are obtained by diagonalizing the electronic Hamiltonian at each QC–environment atomic configuration as (typically) given by MD simulations providing the position of the semiclassical coordinates, with the QC nuclear quantum degrees of freedom either at the corresponding energy-minimized position (when modelling electronic excitations and chemical reactions) or along predefined vibrational modes (when modeling vibrational excitations). Such MD-PMM calculations have been employed to evaluate, beyond the usual electronic spectra [27], the vibrational and vibronic equilibrium spectra [23,28,29,30], the electron-transfer thermodynamics and kinetics [31,32,33,34] and the time-resolved spectroscopic signals [35]. When modeling the vibronic transitions, we assume each vibronic eigenstate as properly described by the Born–Oppenheimer approximation and, hence, given within the coordinate representation by the combination of an electronic eigenstate $\Phi_{i}$ with a corresponding vibrational one $\phi_{i,m}$ (i.e., the mth vibrational eigenstate of the ith electronic eigenstate), providing the i,m vibronic eigenstate via their product $\Phi_{i}\phi_{i,m}$. Different levels of theory can be used to obtain the electronic eigenstates, based on the type of expansion/approximation used to express the perturbation operator within the electronic Hamiltonian [26]. A typically accurate approximation (used in this paper) for the Hamiltonian matrix ${\tilde{H}}_{e}$ representing the electronic Hamiltonian operator ${\hat{H}}_{e}$ within the unperturbed electronic eigenstate basis set $\Phi_{j}^{0}$ (i.e., the eigenstates of the non-interacting QC) is [26]
(1)$$\begin{array}{ccc} {\left[{\tilde{H}}_{e}\right]}_{j,l} & = & \langle \Phi_{j}^{0}|{\hat{H}}_{e}|\Phi_{l}^{0}\rangle ≅\left[U_{j}^{0}+\sum_{N}q_{N,j}^{0}V\left(R_{N}\right)+\Delta V\right]\delta_{j,l}\\ & - & E\left(r_{0}\right)\cdot \langle \Phi_{j}^{0}|\hat{\mu }|\Phi_{l}^{0}\rangle \left(1-\delta_{j,l}\right) \end{array}$$
where $U_{j}^{0}$ is the energy of the *j*th unperturbed electronic eigenstate (i.e, the *j*th unperturbed electronic energy), *N* runs over the QC atoms with positions $R_{N}$ , $q_{N,j}^{0}$ is the *N*th atomic charge in the *j* unperturbed eigenstate, $V(R_{N})$ is the perturbing electric potential at the *N*th atom position, the scalar function ΔV (independent of the electronic states) approximates all the higher order terms of the diagonal elements, $E(r_{0})$ is the perturbing electric field at the reference position $r_{0}$ (typically the QC center of mass), and $\hat{\mu }$ is the dipole operator and $\delta_{j,l}$ is the Kronecker delta. By means of the eigenvectors and eigenvalues of the Hamiltonian matrix provided by Equation (Equation 2), we can obtain any perturbed electronic property at each MD frame, and, in principle, the vibrational eigenstates and frequencies for each perturbed electronic eigenstate. Unfortunately, a complete evaluation of the perturbed vibrational eigenstates to be used when modeling vibronic spectra is computationally unfeasible and, thus, considering that the perturbed vibrational modes are typically virtually identical to the unperturbed ones [36] (i.e., to the vibrational modes of the unperturbed electronic eigenstate best corresponding to the perturbed electronic eigenstate) and the vibrational frequency variations due to the perturbation provide negligible changes in the excitation/relaxation energy of vibronic transitions involving an electronic excitation/relaxation, we can adopt the approximation of using, for each perturbed electronic eigenstate, the unperturbed harmonic vibrational eigenstates and frequencies of the proper unperturbed electronic eigenstate (i.e., the unperturbed electronic eigenstate best matching the perturbed one) to reconstruct the (perturbed) vibronic eigenstates and related properties.

### 2.2. The Excitation and Emission Modeling

From the general expression providing the complete vibronic spectral signal [23], we obtain the extinction coefficient for a given (perturbed) excitation from the vibronic ground eigenstate to the *m* vibrational eigenstate of the ith electronic excited eigenstate (i.e., the $\Phi_{0}\phi_{0,0}\to \Phi_{i}\phi_{i,m}$ vibronic transition) with (perturbed) electronic transition dipole $\langle \Phi_{0}|\hat{\mu }|\Phi_{i}\rangle =\mu_{0,i}$
(2)$$\begin{array}{ccccccccccc} & & & & & & & & \varepsilon_{0,i}^{m}\left(\nu \right) & ≅ & \sum_{\nu_{ref}^{m}}\frac{\Gamma_{A}\left(\nu_{ref}^{m}\right)\;n\left(\nu_{ref}^{m}\right)\;h\nu }{N_{f}}\frac{e^{-\frac{{\left(\nu -\nu_{ref}^{m}\right)}^{2}}{2\sigma^{2}}}}{\sigma \sqrt{2\pi }} \end{array}$$
(3)$$\begin{array}{ccc} \Gamma_{A}\left(\nu_{ref}^{m}\right) & = & \frac{|\mu_{0,i}{|}_{\nu_{ref}^{m}}^{2}}{6\epsilon_{0}c\hbar^{2}}\;{\left|\langle \phi_{0,0}^{0}|\phi_{j_{i},m}^{0}\rangle \right|}_{\nu_{ref}^{m}}^{2} \end{array}$$
and the emission signal $k_{i,0}^{m}\left(\nu \right)\;\rho_{m}\left(\nu \right)$ (with $k_{i,0}^{m}\left(\nu \right)$ the emission rate constant and $\rho_{m}\left(\nu \right)$ the probability density of the emitting molecules) for the (perturbed) vibronic relaxation from the vibrational ground eigenstate of the ith electronic excited eigenstate to the mth vibrational eigenstate of the electronic ground eigenstate (i.e., for the $\Phi_{i}\phi_{i,0}\to \Phi_{0}\phi_{0,m}$ vibronic transition)
(4)$$\begin{array}{ccccccc} & & & & k_{i,0}^{m}\left(\nu \right)\;\rho_{m}\left(\nu \right) & ≅ & \sum_{\nu_{ref}^{m}}\frac{\Gamma_{E}\left(\nu_{ref}^{m}\right)\;n\left(\nu_{ref}^{m}\right)\;{\left(\nu /c\right)}^{3}}{N_{f}}\frac{e^{-\frac{{\left(\nu -\nu_{ref}^{m}\right)}^{2}}{2\sigma^{2}}}}{\sigma \sqrt{2\pi }} \end{array}$$
(5)$$\begin{array}{ccc} \Gamma_{E}\left(\nu_{ref}^{m}\right) & = & \frac{8\pi h|\mu_{0,i}{|}_{\nu_{ref}^{m}}^{2}}{6\epsilon_{0}\hbar^{2}}\;{\left|\langle \phi_{j_{i},0}^{0}|\phi_{0,m}^{0}\rangle \right|}_{\nu_{ref}^{m}}^{2} \end{array}$$
where *c* is the light speed, $\epsilon_{0}$ is the vacuum dielectric constant, *h* is the Planck constant (ℏ=h/(2π)), the summation runs over the frequency bins used to construct the spectrum and identified by the mth vibronic transition reference frequencies $\nu_{ref}^{m}$ (i.e., the bin middle values), $N_{f}$ is the total number of MD frames (necessarily a huge number to obtain a reliable spectral signal), $n(\nu_{ref}^{m})$ is the number of MD frames with the mth vibronic transition frequency ν within the bin centered in $\nu_{ref}^{m}$, and $|\mu_{0,i}{|}_{\nu_{ref}^{m}}^{2}$ is the corresponding electronic vertical transition dipole mean square norm as obtained averaging over the bin MD frames. Both the transition frequency and transition electronic dipole at each MD frame can be provided by the PMM using the MD frame semiclassical configuration to obtain the environment perturbation and QC atomic positions, relaxing only the QC vibrational quantum and semiclassical degrees of freedom at their energy-minimized positions [23,27], as given by the minimum energy structure of the unperturbed electronic ground eigenstate (Equations (Equation 2)–(Equation 3)) or the $j_{i}$ unperturbed electronic eigenstate best corresponding at each frame to the ith perturbed electronic eigenstate (Equations (Equation 4)–(Equation 5)). Moreover, $|\langle \phi_{0,0}^{0}|\phi_{j_{i},m}^{0}\rangle {|}_{\nu_{ref}^{m}}^{2}$ and $|\langle \phi_{j_{i},0}^{0}|\phi_{0,m}^{0}\rangle {|}_{\nu_{ref}^{m}}^{2}$ are the mean square norms of the unperturbed vibrational eigenstate overlap, as obtained by averaging the bin frames possibly corresponding to different conformational basins, each providing different unperturbed properties. Furthermore, $\phi_{0,0}^{0}$ is the vibrational ground eigenstate of the unperturbed electronic ground eigenstate (typically virtually identical to the perturbed one), $\phi_{j_{i},0}^{0}$ is the vibrational ground eigenstate of the unperturbed $j_{i}$ electronic eigenstate, $\phi_{j_{i},m}^{0}$ is the mth vibrational eigenstate of the $j_{i}$ unperturbed electronic eigenstate, $\phi_{0,m}^{0}$ is the mth vibrational eigenstate of the unperturbed electronic ground eigenstate and they are all modeled as harmonic vibrational eigenstates, each as obtained at the corresponding electronic eigenstate energy minimum for the vibrational coordinates. Note that for the excitation processes, the MD simulations used must correspond to the perturbed electronic ground eigenstate ensemble virtually identical to the unperturbed electronic eigenstate one, while for the relaxation processes the ensemble of the perturbed excited electronic eigenstate required is possibly not corresponding to a single unperturbed electronic eigenstate ensemble. Finally, the Gaussian distributions centered in $\nu_{ref}^{m}$ approximate, beyond the homogeneous broadening typically negligible in electronic excitations/relaxations, the broadening due to the QC semiclassical vibrations disregarded in PMM calculations, with the variance possibly estimated by a set of unperturbed electronic excitation energies corresponding to a set of QC configurations as obtained in principle by the MD sampling of the QC semiclassical vibrations [23] (note that, for sake of simplicity, we assume the same variance for all the Gaussian distributions).

In practice, for each electronic excitation by extracting the MD frames of the ground-state simulation with the ith perturbed electronic excited eigenstate best corresponding to a given jth unperturbed one and, for a flexible QC, also corresponding to a single conformation (MD sub-ensemble), we obtain the corresponding excitation vertical electronic spectrum by means of the electronic vertical energy and transition dipole, as provided by PMM [27], via
(6)$$\begin{array}{ccccccccccccccccccc} & & & & & & & & & & & & & & & & \varepsilon_{0,i}\left(\nu \right) & ≅ & \sum_{\nu_{ref}}\frac{\Gamma_{A}\left(\nu_{ref}\right)\;n\left(\nu_{ref}\right)\;h\nu }{N_{f}}\frac{e^{-\frac{{\left(\nu -\nu_{ref}\right)}^{2}}{2\sigma^{2}}}}{\sigma \sqrt{2\pi }} \end{array}$$
(7)$$\begin{array}{ccc} \Gamma_{A}\left(\nu_{ref}\right) & = & \frac{|\mu_{0,i}{|}_{\nu_{ref}}^{2}}{6\epsilon_{0}c\hbar^{2}} \end{array}$$
with, now, the summation running over the frequency bins identified by the vertical electronic transition reference frequencies $\nu_{ref}$, $n(\nu_{ref})$ the number of MD frames with the vertical electronic transition frequency ν within the bin centered in $\nu_{ref}$ and $|\mu_{0,i}{|}_{\nu_{ref}}^{2}$ the corresponding electronic vertical-transition dipole mean square norm as obtained by averaging over the bin MD frames. Note that by conformation we mean a configurational region of the internal semiclassical coordinate space (typically corresponding to a harmonic or quasi-harmonic basin) where all the QC electronic vertical-transition properties, except the vertical-transition energy, can be obtained at a single ground-state reference structure corresponding to a local energy minimum of the unperturbed electronic ground eigenstate (i.e., all such electronic properties are independent of the local configurational changes), considering the inherent fluctuations in the semiclassical internal coordinates as local classical vibrations. On the basis of the assumed negligible perturbation effects on the vibrational eigenstates and eigenvalues and noting that within a single sub-ensemble the unperturbed $j_{i}$ electronic eigenstate is invariant and, hence, we can use the $j_{i}\to j$ substitution, we can reconstruct for each sub-ensemble the mth vibronic spectral peak by multiplying the vertical electronic spectrum by $|\langle \phi_{0,0}^{0}|\phi_{j,m}^{0}\rangle {|}^{2}$ (the square norm of the unperturbed vibrational eigenstate overlap in a single sub-ensemble, fully constant over those MD frames) and locating its maximum at
(8)$$\nu_{m}=\nu_{m}^{0}+\nu_{el}-\nu_{el}^{0}$$
where $\nu_{m}^{0}$ is the excitation frequency of the unperturbed mth vibronic transition (unperturbed mth vibronic frequency) possibly corrected to match the experimental gas-phase value, $\nu_{el}$ is the (perturbed) excitation frequency of the maximum of the electronic vertical spectrum and $\nu_{el}^{0}$ is the excitation frequency of the unperturbed vertical electronic transition (unperturbed electronic vertical frequency). The sum of these vibronic peaks provides the sub-ensemble vibronic spectrum and, thus, summing such spectra over all the sub-ensembles, each statistically weighted by the corresponding probability as provided by the ground state MD simulation, and then over all the relevant electronic excitations, we obtain the complete vibronic excitation spectrum. When dealing with electronic relaxations, a similar procedure is used, except that we utilize the excited-state MD simulation instead of the ground-state one to obtain the ensemble statistics, with the following expression
(9)$$\begin{array}{ccccccccccccccc} & & & & & & & & & & & & k_{i,0}\left(\nu \right)\;\rho \left(\nu \right) & ≅ & \sum_{\nu_{ref}}\frac{\Gamma_{E}\left(\nu_{ref}\right)\;n\left(\nu_{ref}\right)\;{\left(\nu /c\right)}^{3}}{N_{f}}\frac{e^{-\frac{{\left(\nu -\nu_{ref}\right)}^{2}}{2\sigma^{2}}}}{\sigma \sqrt{2\pi }} \end{array}$$
(10)$$\begin{array}{ccc} \Gamma_{E}\left(\nu_{ref}\right) & = & \frac{8\pi h|\mu_{0,i}{|}_{\nu_{ref}}^{2}}{6\epsilon_{0}\hbar^{2}} \end{array}$$
providing for a given sub-ensemble the emission vertical electronic signal to be multiplied by $|\langle \phi_{j,0}^{0}|\phi_{0,m}^{0}\rangle {|}^{2}$, with now the electronic vertical transition frequency and transition dipole as obtained at the excited reference structure (i.e., the local energy minimum of the proper unperturbed electronic excited eigenstate). Moreover, we need for each electronic transition as many independent excited-state MD simulations as all the electronic conditions accessible to the perturbed excited eigenstate of interest, each well corresponding to a specific unperturbed electronic eigenstate providing the QC atomic charges and intramolecular properties to be used in the MD force field. Moreover, for each of such MD simulations, the sub-ensembles can only correspond to the different conformational conditions, as the final state of the emission is the electronic ground eigenstate virtually always well corresponding to the unperturbed electronic ground eigenstate (note that we need to remove from the excited-state simulation ensemble, if present, those configurations where the perturbed electronic excited eigenstate of interest is not coherent with the unperturbed one used to define the force field).

Finally, it is worth remarking that for each conformation of the ensemble MD simulation (the ground-state simulation for the excitation process and the excited-state simulation for the emission process), we use the corresponding ground- and excited-state reference structures, i.e., the energy minima of the unperturbed ground and excited electronic eigenstates, to obtain the unperturbed normal modes defining for each electronic eigenstate the QC quantum vibrational degrees of freedom via the modes with frequencies such that $h\nu >k_{B}T$ ($k_{B}$ is the Boltzman constant and *T* is the absolute temperature) and the QC semiclassical internal coordinates via all the other modes, except the roto-translational ones corresponding to the QC spatial changes due to the differential variations in the roto-translational coordinates (we always consider the roto-translational coordinates given by the center-of-mass Cartesian position and the Eulerian angles as semiclassical degrees of freedom). In the appendix, we discuss the conditions ensuring that such a choice provides for each conformation the proper definition of the internal coordinates.

### 2.3. A Simplified Efficient Strategy to Evaluate the Vibrational Overlap

From the previous theory subsections, it is evident that from the diagonalization of the perturbed electronic Hamiltonian matrix we can, in principle, obtain at each MD frame all the information necessary to reconstruct the spectroscopic signal according to Equations (Equation 6)–(Equation 10), once we have a proper estimate of $|\langle \phi_{0,0}^{0}|\phi_{j,m}^{0}\rangle {|}^{2}$ and $|\langle \phi_{j,0}^{0}|\phi_{0,m}^{0}\rangle {|}^{2}$ for each vibronic excitation and relaxation, respectively. However, the explicit evaluation of such vibrational eigenstate overlaps for the typically huge number of possibly significant vibronic transitions is computationally very demanding. Therefore, the available accurate methods for such calculations can be used for a limited number of vibronic peaks (a few hundred) and usually require for each electronic eigenstate the energy minimum in full configurational space [23,37]. Such restrictions can be overcome using proper approximations leading to a relevantly simplified procedure. In fact, when assuming for different electronic eigenstates, within each MD sub-ensemble, the corresponding quantum vibrational modes with approximately the same frequency being virtually identical, i.e., they are defined by the same mass-weighted Hessian eigenvector and, thus, share the same mode coordinate with only different minimum energy positions, we can express the vibrational eigenstate overlaps via the product of the single-mode eigenstate overlaps $\langle \eta_{j,n,k}^{0}\left(\beta_{n}\right)|\eta_{j^{\prime },n,k^{\prime }}^{0}\left(\beta_{n}\right)\rangle$ where $\eta_{j,n,k}^{0}\left(\beta_{n}\right)$ and $\eta_{j^{\prime },n,k^{\prime }}^{0}\left(\beta_{n}\right)$ are the *k* and $k^{\prime }$ vibrational eigenstates of the *n* quantum mode, with coordinate $\beta_{n}$, for the *j* and $j^{\prime }$ electronic eigenstates, respectively. Note that within our assumption, $\eta_{j,n,k}^{0}\left(\beta_{n}\right)$ and $\eta_{j^{\prime },n,k}^{0}\left(\beta_{n}\right)$ are identical except for the *j* and $j^{\prime }$ electronic eigenstate minimum-energy position along the mode coordinate $\beta_{n}$. Therefore, for the $j,j^{\prime }$ electronic eigenstate couple, the vibrational ground–ground overlap (i.e., the m=0 vibronic transition) is
(11)$$\langle \phi_{j,0}^{0}\left(\beta \right)|\phi_{j^{\prime },0}^{0}\left(\beta \right)\rangle ≅\Pi_{n}\langle \eta_{j,n,0}^{0}\left(\beta_{n}\right)|\eta_{j^{\prime },n,0}^{0}\left(\beta_{n}\right)\rangle$$
and for the *m* vibronic transition involving for the $j^{\prime }$ electronic eigenstate the $k_{m}$ excited vibrational eigenstate of the $n_{m}$ mode (i.e., single mode excitation), we have
(12)$$\begin{array}{ccc} & & \langle \phi_{j,0}^{0}\left(\beta \right)|\phi_{j^{\prime },m}^{0}\left(\beta \right)\rangle ≅\\ & & \langle \eta_{j,n_{m},0}^{0}\left(\beta_{n}\right)|\eta_{j^{\prime },n_{m},k_{m}}^{0}\left(\beta_{n_{m}}\right)\rangle \;\Pi_{n\neq n_{m}}\langle \eta_{j,n,0}^{0}\left(\beta_{n}\right)|\eta_{j^{\prime },n,0}^{0}\left(\beta_{n}\right)\rangle \end{array}$$
with $\beta =\{\beta_{1},\beta_{2},...,\beta_{n},...\}$ the vector representing all the quantum-mode coordinates. When considering vibronic transitions involving multiple mode excitations for the $j^{\prime }$ electronic eigenstate (i.e., the $k_{m,l}$ excited vibrational eigenstates of the $n_{m,l}$ quantum modes with coordinates $\beta_{n_{m,l}}$), we have the general overlap expression
(13)$$\begin{array}{ccc} & & \langle \phi_{j,0}^{0}\left(\beta \right)|\phi_{j^{\prime },m}^{0}\left(\beta \right)\rangle ≅\\ & & \Pi_{l}\langle \eta_{j,n_{m,l},0}^{0}\left(\beta_{n_{m,l}}\right)|\eta_{j^{\prime },n_{m,l},k_{m,l}}^{0}\left(\beta_{n_{m,l}}\right)\rangle \;\Pi_{n\neq n_{m}}\langle \eta_{j,n,0}^{0}\left(\beta_{n}\right)|\eta_{j^{\prime },n,0}^{0}\left(\beta_{n}\right)\rangle \end{array}$$
with $n_{m}=\left\{n_{m,1},n_{m,2},n_{m,3},...,n_{m,l},...\right\}$ the vector representing all the excited quantum modes. Finally, note that the vibrational eigenstates of any electronic state form a complete basis set for the vibrational state space, thus necessarily implying
(14)$$\sum_{m}{\left|\langle \phi_{j,0}^{0}|\phi_{j^{\prime },m}^{0}\rangle \right|}^{2}=1$$

with *m* running over all the vibrational eigenstates of the unperturbed $j^{\prime }$ electronic state.

Equations (Equation 11)–(Equation 13) clearly show that within the approximation used, the evaluation of the vibrational overlap involving a very complex multiple integral is reduced to the product of single-mode integrals, each trivially evaluated using two harmonic vibrational wavefunctions corresponding to identical modes except for the minimum energy position along the shared mode coordinate. This approach can then allow us to include in the spectrum model a huge number of vibronic transitions, making straightforward the separation between the quantum vibrational coordinates and the semiclassical ones, thus possibly avoiding the restriction of using ground and excited reference structures strictly corresponding to full-space energy minima (we need only the quantum coordinates to be properly energy minimized).

## 3. Application to Aqueous Indole

### 3.1. Computational Details

The geometries of indole in the ground state and first three excited states were optimized in vacuum by means of CCSD and EOM-CCSD, using 6-311+G(d) as the basis set. The gas-phase electronic properties of the first six excited states were calculated for each geometry using EOM-CCSD, with 6-311+G(d) as the basis set. The harmonic vibrational eigenstates and frequencies were calculated by means of DFT and TD-DFT, with the functional M06-2X [38] and 6-311+G(d) as the basis set (the benchmarking [38,39,40,41,42,43] which resulted in this choice is described in detail in the Appendix A; see Appendix A and Appendix A), using both the geometries optimized with the same level of theory (DFT) as well as the geometries optimized with CCSD and EOM-CCSD. The geometry optimizations and the vibrational-frequencies calculations were performed using Gaussian 16 [44], while the electronic properties of the ground state and the excited states were calculated using Q-Chem [45]. The calculated gas-phase vibronic transition energies were corrected in order to match the 0–0 transition with the experimental gas-phase value. This resulted in a shift in the vibronic transitions associated with the excitation to L${}_{b}$ of −0.32 eV (arising from the difference between the reported gas-state experimental value of the 0-0 transition of 4.37 eV [8] and our calculated unperturbed value of 4.69 eV provided by CCSD/6-311+G(d) calculations) and a shift by −0.57 eV for the vibronic transitions associated with the excitation to L${}_{a}$ (given by the reported gas-state experimental value of the 0–0 transition of 4.54 eV [8] and our calculated unperturbed value of 5.11 eV). Finally, the value of σ used to construct the spectra (0.0004 frequency atomic units corresponding to 0.068 eV) was obtained by tuning the value estimated for Pyrene (about 0.02 eV) in a previous work [23] in order to optimize the spectral lineshape. The Pyrene value was evaluated by calculating the variance in the electronic transition energies of the semi-rigid chromophore in gas phase for several configurations within its MD conformational basin (harmonic-like basin).

NVT MD simulations of indole and 949 TIP3P water molecules were carried out in a cubic box of side 3.02356 nm using periodic boundary conditions, at 300 K. The size of the box was chosen in order to reproduce the experimental density of the system [46]. CHARMM36 [47,48] was used as the force field and the canonical sampling was achieved using the velocity rescaling thermostat [49]. Indole was considered in the ground or in one of the spectroscopically active (unperturbed) electronic excited states (the L${}_{a}$ and L${}_{b}$ excited states) by replacing the values of its atomic charges in the force field with the ESP charges [50,51] calculated at the EOM-CCSD/6-311+G(d) level. For each system simulation, the MD trajectory was carried out for 400 ns with a 2 fs timestep, which resulted in 20,000 frames (taken each 20 ps) to be used in the MD-PMM procedure. The MD simulations were performed using Gromacs 2020.1 [52].

### 3.2. Results

Indole (see Figure 1) is a semi-rigid molecule, thus allowing us to consider a single conformational (quasi-harmonic) basin to fully describe its spectroscopic behavior. According to the model described in the theory section, we used as the reference structure for each perturbed electronic eigenstate of interest the energy minimum of the unperturbed electronic eigenstate best corresponding to the perturbed one, hence providing all the unperturbed electronic properties and vibrational modes and frequencies to be used. In Figure 2, we show the matrices given by the squared inner products of the mass-weighted Hessian eigenvectors (the squared elements of the Duschinsky matrix), as obtained for the unperturbed QC at the energy minima of the ground and L${}_{b}$ (panel A), or the ground, and L${}_{a}$ (panel B), the electronic, states. From the figure, it is clear that such matrices are nearly diagonal, well matching the requirement for the use of our simplified treatment of the vibronic transitions (note that, in both matrices, modes with the same index always have nearly identical frequencies).

Concerning the modeling of the absorption spectrum, we obtained the ground-state ensemble by performing a long equilibrium MD simulation (400 ns) with the indole force-field parameters corresponding to the QC unperturbed electronic ground-state charge distribution (as usual, the QC perturbed electronic ground state is virtually identical to the unperturbed one, data not shown). From the obtained ground-state MD ensemble, via PMM, we reconstructed the corresponding distributions of the projections (Hermitian products) of the first and second perturbed electronic excited states on the unperturbed electronic states [25] (the ground state GS, the excited states L${}_{b}$ L${}_{a}$ and the dark state $\pi \sigma^{*}$), obtained at the (unperturbed) ground-state optimized geometry (ground-state geometry). From such distributions, shown in Figure 3 and Figure 4, it is evident that within the ground-state MD ensemble the perturbed first electronic excited state mostly corresponds to the L${}_{b}$ electronic state, while the perturbed second electronic excited state is largely corresponding to the L${}_{a}$ electronic state (see panels A). Interestingly, from the panel B of the same figures, we can realize that both perturbed electronic excited states essentially fluctuate between the L${}_{a}$ and L${}_{b}$ states, being nearly identical to only one of them at each MD frame (we disregard the few MD frames with the perturbed second electronic excited state projecting onto the dark state $\pi \sigma^{*}$).

In order to obtain the absorption spectrum (including the vibronic details), we extracted from the ground-state MD simulation, for each of the first two perturbed electronic excited states, the MD frames where the perturbed electronic excited state of interest (as obtained at the ground-state geometry) is virtually identical either to the L${}_{b}$ or to the L${}_{a}$ state, thus allowing us to reconstruct (see the theory section) the corresponding vibronic spectra (for each MD frame we assumed the same electronic-state assignment over the whole relevant range of the mode coordinates). The complete absorption spectrum as shown in Figure 5 was obtained from the weighted summation of all the contributions (including the vibronic details) due to both the excited states. In this figure, we compare the experimental absorption spectrum with the calculated ones, as obtained according to the theory section using either our vibronic treatment or the vibronic structure provided by the algorithms implemented in Gaussian [37] (Gaussian vibronic treatment). Note that, due to the limited number of vibronic transitions considered by the Gaussian vibronic treatment, in order to mimic in the corresponding spectral signal the effect of the missing vibronic transitions, we scaled the estimated squared vibrational-state overlaps (as provided by Gaussian via the unperturbed absorption intensities) to normalize their sum (see Equation (Equation 14)): we assume all the missing vibronic transitions as included within the absorption range of interest and distributed to provide the same intensity relative increment for each vibronic transition considered. Therefore, the vibronic spectrum based on the Gaussian vibronic treatment reported should be considered as corresponding to the highest possible absorption intensity: i.e., the intensity of the spectrum we would obtain including within the Gaussian vibronic treatment all the relevant vibronic transitions, with, hence, no normalization scaling of the squared vibrational state overlaps, is likely to be lower. From the figure, it is clear the higher accuracy of our vibronic treatment in reproducing the details of the spectrum, although the Gaussian vibronic treatment also provides a proper spectrum intensity and width (both calculated spectra properly reproduce the position of the maximum). It is worth noting that due to computational limitations we always had to employ DFT calculations for evaluating the mass-weighted Hessian at the reference structures (electronic-energy minima) used to obtain the vibronic structure for the first two electronic transitions (the energy minima of the GS, L${}_{a}$ and L${}_{b}$ unperturbed electronic eigenstates). Therefore, the use of the vibronic treatment implemented in Gaussian, requiring full-space minima, forced us to also employ DFT calculations to obtain the optimized structures to be used for the Hessian calculations (for the PMM calculations we always used the higher level CCSD and EOM-CCSD methods). Conversely, when using our (simplified) vibronic treatment, requiring a minimum only within the quantum-mode subspace, we were able to utilize DFT Hessian calculations for reference structures optimized at CCSD and EOM-CCSD level, thus possibly guaranteeing a higher accuracy. In addition, our approach allowed us to include a huge number of vibronic transitions (over a million transitions instead of about 100, as used within the Gaussian vibronic treatment) providing a virtually complete reconstruction of the vibronic structure for each electronic excitation. The results reported in Figure 5, indicating that the approximations we used to simplify the vibronic structure calculations are accurate, plainly show that the inclusion of a very large number of vibronic transitions and the possibility to treat separately the quantum- and classical-mode subspaces can provide a significant improvement in modeling vibronic spectra.

In modeling the emission process, we had to consider the proper excited-state MD ensemble to be used for reconstructing the fluorescence spectrum due to the first excited-state radiative relaxation (as usual, we assumed no other radiative relaxations as a consequence of the much faster non-radiative transitions from higher excited states to the first excited state). From the absorption data discussed above, we know that the first two perturbed electronic excited states can be conceived as fluctuating between the L${}_{a}$ and L${}_{b}$ states, thus suggesting that the emission process should be modeled using the corresponding excited-state MD ensembles and geometries, i.e., using the corresponding excited-state MD force field and optimized geometry (L${}_{a}$ and L${}_{b}$ geometries). Note that as the fluorescence mean lifetime is much longer than the ones of the inter-conversion processes involving the first three perturbed excited states (within the GS geometry and MD ensemble, basically corresponding to the L${}_{b}$, L${}_{a}$ and $\pi \sigma^{*}$ unperturbed states, respectively), we cannot assume the L${}_{b}$ ensemble as the proper excited-state MD ensemble to be used. Therefore, we used two different excited-state MD ensembles and geometries, using the L${}_{b}$ and L${}_{a}$ optimized geometries and corresponding MD simulations (we disregarded the spectroscopically inactive dark state). In order to verify the consistency of the MD ensembles obtained, we evaluated the mean-squared projections (mean-squared Hermitian products) of the perturbed electronic first excited state over the first four unperturbed electronic states (the ground state GS, the excited states L${}_{b}$, L${}_{a}$ and the dark state $\pi \sigma^{*}$), as obtained either at the L${}_{b}$ or L${}_{a}$ geometry and MD ensemble (see panels A of Figure 6 and Figure 7). From these figures, it is evident that within the L${}_{b}$ geometry and MD ensemble the perturbed first electronic state is coinciding with the L${}_{b}$ state, while within the L${}_{a}$ geometry and MD ensemble the same perturbed eigenstate is largely corresponding to the L${}_{a}$ state, although still projecting onto the L${}_{b}$ state. Interestingly, from panel B of Figure 7 it emerges that in this last case the perturbed first electronic excited state is not fluctuating between the L${}_{a}$ and L${}_{b}$ states but it is, rather, a stationary combination of them. These results indicate that we can use in both cases the complete MD simulation to obtain a consistent excited-state MD ensemble for evaluating the emission spectrum (for the L${}_{a}$-state MD ensemble considering the ≈ 80 % projection of the perturbed first excited state onto the L${}_{a}$ state).

Finally, in Figure 8 we show the comparison between the calculated emission spectra, as obtained employing either our vibronic treatment or the Gaussian vibronic treatment, with the experimental emission spectrum for the L${}_{a}$ (panel A) and L${}_{b}$ (panel B) excited-state geometries and MD ensembles. From this figure, it is evident that only the L${}_{a}$ geometry and MD ensemble reproduce the experimental signal, with a clear improvement in reproducing the spectrum when using our vibronic treatment. The comparison of such results with those obtained for the absorption process shows that after the excitation process, a fast relaxation must occur, providing a dramatic change in the perturbed first electronic excited-state population from being largely coinciding with the L${}_{b}$ state (see Figure 3) to largely resembling the L${}_{a}$ state.

## 4. Conclusions

The use of general and typically accurate approximations for modelling vibronic transitions (see the Theory section), allowed us to relevantly simplify the vibronic treatment [16,17], resulting in a very efficient and accurate reconstruction of the absorption and emission spectra, within the MD-PMM framework. In fact, taking advantage of the approximately identical vibrational modes and frequencies of the energy minima involved in the electronic transitions, we are able to explicitly include in our treatment a huge number of vibronic transitions, which are necessary to reproduce the details of the spectral signal. Moreover, stemming from the approximations/simplifications adopted, we can easily discriminate between classical-like and quantum-mode vibronic transitions, allowing us to uncouple the corresponding vibronic calculations, hence permitting the use of the quantum-mode transitions only and avoiding the restriction of using ground and excited reference structures strictly corresponding to full-space energy minima, i.e., we need only the quantum coordinates to be properly energy minimized. Such a feature of the method allows the use of high-level calculations for the evaluation of properly minimized reference configurations and related fully reliable properties (e.g., by means of EOM-CCSD), still calculating the corresponding harmonic vibrational modes and frequencies via a less accurate but computationally much more affordable method (e.g., TD-DFT), often unable to provide the correct minimum energy position for the classical-like modes. The application of such an approach to aqueous Indole quantitatively reproduces both the experimental absorption and emission spectra with higher accuracy than the one obtained by employing the usual vibronic treatment as implemented in Gaussian [16,37]. Moreover, it also provided indications on indole-complex excited-state behavior, shedding light on the different sub-populations accessed by means of the absorption process, i.e., the L${}_{a}$, L${}_{b}$ and $\pi \sigma^{*}$ (unperturbed) state sub-populations of the perturbed first and second electronic excited states. Finally, the comparison of the calculated emission spectra as obtained by the first (perturbed) excited-state MD ensemble and geometry corresponding to either the L${}_{a}$ or the L${}_{b}$ states clearly showed that, although the excitation provides a first perturbed excited state largely corresponding to the L${}_{b}$ state, in the emission, the L${}_{a}$ ensemble and geometry can only provide a proper spectral signal, thus showing that a fast relevant relaxation of the electronic perturbed first excited-state population (converting the L${}_{b}$ sub-population into the L${}_{a}$ one) must occur.

## Figures and Tables

**Figure 1 molecules-27-08135-f001:**
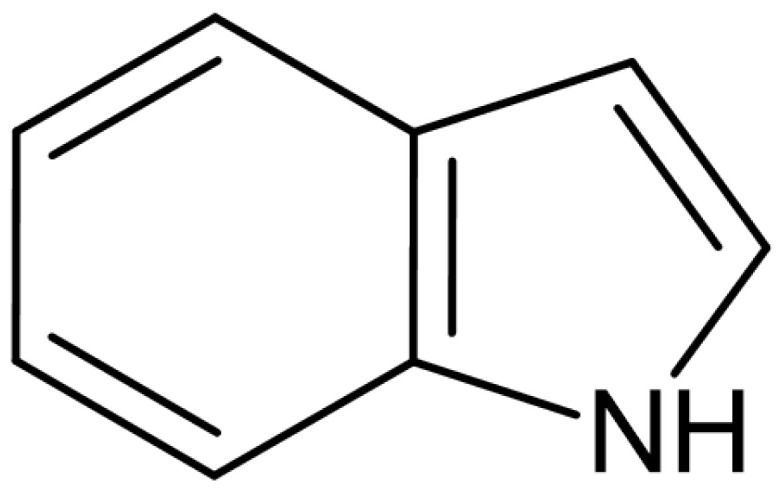
Indole structure.

**Figure 2 molecules-27-08135-f002:**
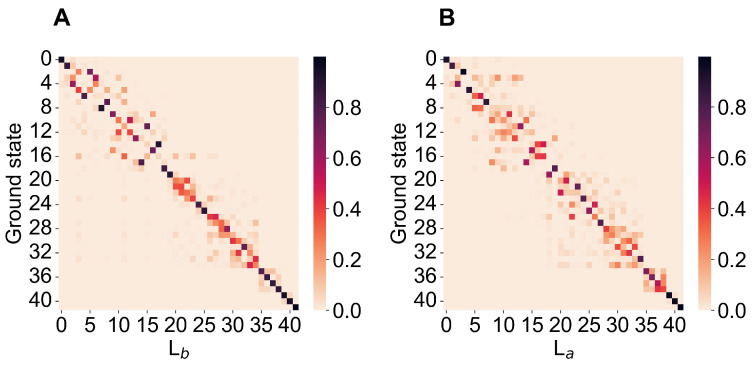
Matrices obtained by the squared elements of the Duschinsky matrix associated to the ground-to-L${}_{b}$-state transition (**A**) and to the ground-to-L${}_{a}$-state transition (**B**). Note that, in both matrices, modes with the same index always have nearly identical frequencies.

**Figure 3 molecules-27-08135-f003:**
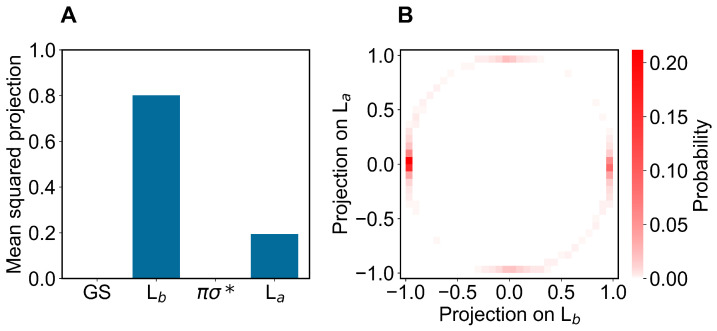
(**A**): Mean-squared projections of the perturbed first electronic excited state of indole, on the first four unperturbed electronic eigenstates (the ground state GS, the excited states L${}_{b}$, L${}_{a}$ and the dark state $\pi \sigma^{*}$, in the figure ordered according to increasing energy) as obtained at the (unperturbed) ground-state geometry by the ground-state MD ensemble. (**B**): Probability distribution of the pair values of the projections of the perturbed first electronic excited state of indole, on the L${}_{b}$ and L${}_{a}$ states as obtained at the (unperturbed) ground-state geometry by the ground-state MD ensemble.

**Figure 4 molecules-27-08135-f004:**
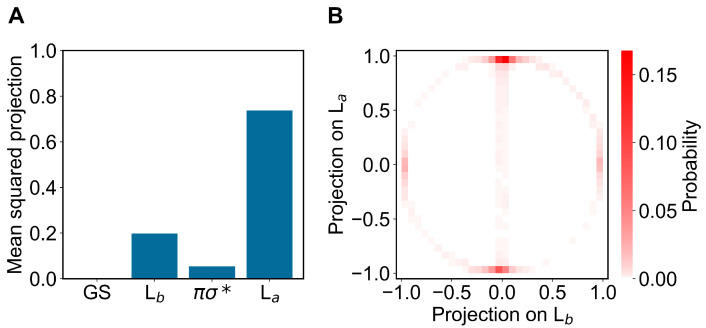
(**A**): Mean-squared projections of the perturbed second electronic excited state of indole, on the first four unperturbed electronic eigenstates (the ground state GS, the excited states L${}_{b}$, L${}_{a}$ and the dark state $\pi \sigma^{*}$, in the figure ordered according to increasing energy) as obtained at the (unperturbed) ground-state geometry by the ground-state MD ensemble. (**B**): Probability distribution of the pair values of the projections of the perturbed second electronic excited state of indole, on the L${}_{b}$ and L${}_{a}$ states as obtained at the (unperturbed) ground-state geometry by the ground-state MD ensemble.

**Figure 5 molecules-27-08135-f005:**
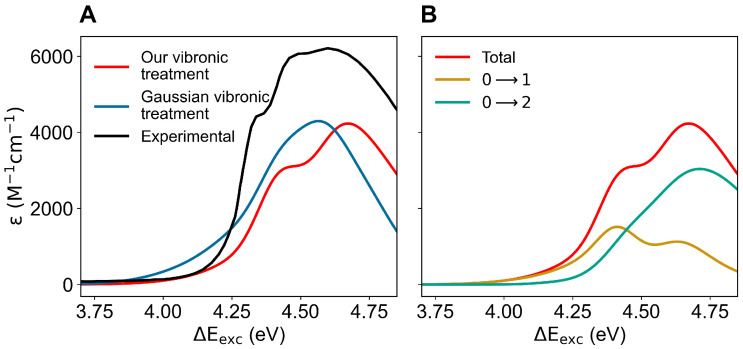
(**A**): Calculated (colored) and measured (black; adapted from Hilaire et al. [53]) absorption spectra of indole in aqueous solution. The predicted spectra were obtained using our vibronic treatment (red) and with the vibronic structure calculated using the algorithms implemented in Gaussian (blue), both as provided by the ground-state MD ensemble. (**B**): Calculated total absorption spectrum (using our vibronic treatment) of indole in aqueous solution and the corresponding contributions to the vibronic spectrum due to the first two electronic transitions.

**Figure 6 molecules-27-08135-f006:**
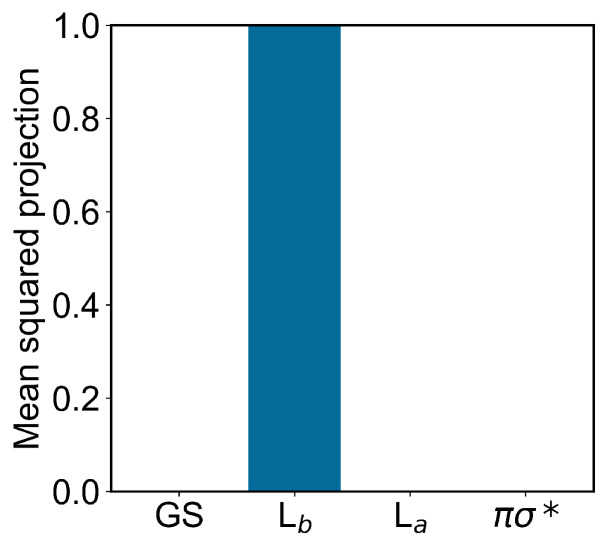
Mean-squared projections of the perturbed first electronic excited state of indole, on the first four unperturbed electronic eigenstates (the ground state GS, the excited states L${}_{b}$, L${}_{a}$ and the dark state $\pi \sigma^{*}$, in the figure ordered according to increasing energy) as obtained at the L${}_{b}$ geometry by the L${}_{b}$-state MD ensemble.

**Figure 7 molecules-27-08135-f007:**
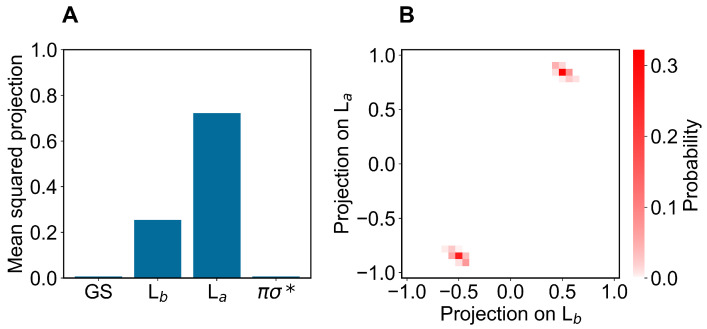
(**A**): Mean-squared projections of the perturbed first electronic excited state of indole, on the first four unperturbed electronic eigenstates (the ground state GS, the excited states L${}_{b}$, L${}_{a}$ and the dark state $\pi \sigma^{*}$, in the figure ordered according to increasing energy) as obtained at the L${}_{a}$ geometry by the L${}_{a}$ state MD ensemble. (**B**): Probability distribution of the pair values of the projections of the perturbed first electronic excited state of indole, on the L${}_{b}$ and L${}_{a}$ states as obtained at the L${}_{a}$ geometry by the L${}_{a}$-state MD ensemble.

**Figure 8 molecules-27-08135-f008:**
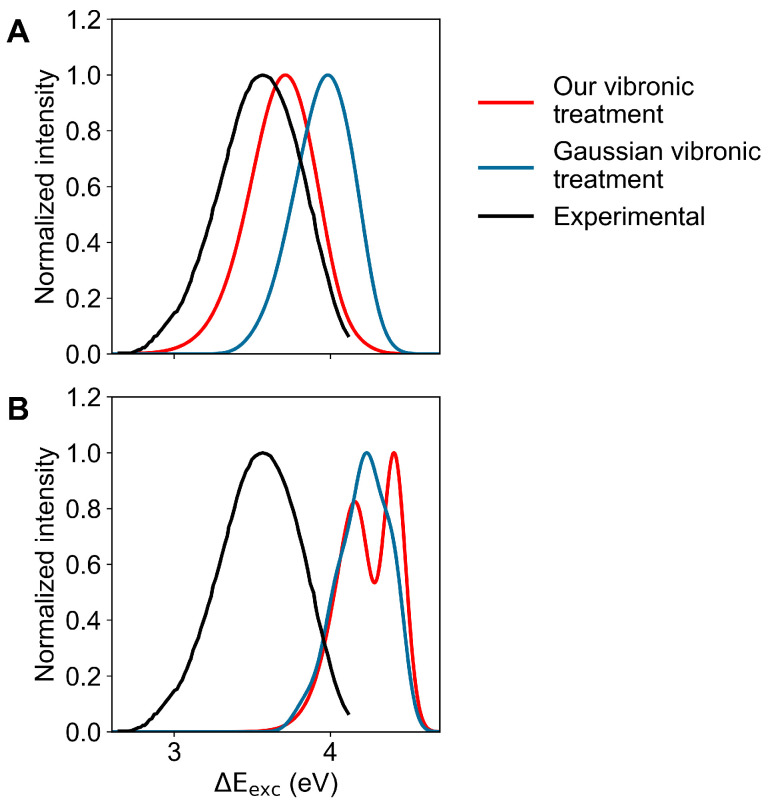
Calculated (coloured) and measured (black; adapted from Hilaire et al. [53]) normalized emission spectra of indole in aqueous solution. Spectra calculated using our vibronic treatment are reported in red, while the spectra obtained by the vibronic structure calculated using the algorithms implemented in Gaussian are reported in blue. (**A**): Spectra calculated in the L${}_{a}$-state MD ensemble. (**B**): Spectra calculated in the L${}_{b}$-state MD ensemble.

## Data Availability

The data presented in this study are available on request from the corresponding author.

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
