# Peer review of "A Simplified Treatment for Efficiently Modeling the Spectral Signal of Vibronic Transitions: Application to Aqueous Indole"

_molecules, 2022, doi:10.3390/molecules27238135_

Round 1

Reviewer 1 Report

The present manuscript describes a simplified approach for modeling of the vibronic structures of electronic absoption and fluorescence spectra with the aid of quantum chemical calculations and MD simulations. The proposed approach was applied for the spectra of indole in water solution. The paper is very well written and the authors are clearly experts in their field. All procedures are well described and scientifically sound. The conclusions are significant and fully supported by the data that they assembled. There really is nothing that I have to criticize and hence I support publication in the present form. 
One very minor comment - Was there any difference between the CCSD and CCSD(T) geometry optimizations of the title molecule? The latter are more robust in the thermochemical calculations.

Author Response

 The present manuscript describes a simplified approach for modeling of the vibronic
structures of electronic absoption and fluorescence spectra with the aid of quantum chemical
calculations and MD simulations. The proposed approach was applied for the spectra of
indole in water solution. The paper is very well written and the authors are clearly experts in
their field. All procedures are well described and scientifically sound. The conclusions are
significant and fully supported by the data that they assembled. There really is nothing that I
have to criticize and hence I support publication in the present form.
One very minor comment - Was there any difference between the CCSD and CCSD(T)
geometry optimizations of the title molecule? The latter are more robust in the
thermochemical calculations.

We thank the referee for the positive feedback. Unfortunately, the use of CCSD(T) to perform
the geometry optimization is beyond the computational resources currently at our disposal,
especially when the optimization of the Lb and La excited states is concerned (requiring the
even more expensive EOM-CCSD(T)). We felt confident in the choice of this method
considering its successful application in previous works in the literature on the same system,
such as the recent one by Abou-Hatab and Matsika (J. Phys. Chem. B 2019, 123, 34,
7424–7435)

Reviewer 2 Report

In this work, the authors have developed a practical computational method to simulate absorption and emission spectra including the significant contribution of vibronic transitions for large molecules in aqueous solutions. In this manuscript, it is shown that the proposed method works well for aqueous indole by comparing with the corresponding experimental data. Basically, this is an interesting study and I believe that the manuscript can be published once the following technical (minor) points.

1.       Page 8, L299-300: The authors have stated that the 0-0 transition is corrected. More detailed descriptions should be given.

2.       I cannot find the physical definition of “Projection” presented in Figures 3, 4 and 7.

3.       I cannot find the actual values for sigma of eqs. (2), (4), (6) and (9). I would like to know how to determine appropriate values for sigma.

4.       The CCSD and EOM-CCSD methods are known to be gold standard in the quantum chemistry field even if a relatively small basis set (6-311+G(d)) is used. However, I think that the computed spectra may depend on the DFT functional used (M06-2X). Is it true? Or the result does not depend on the DFT functional. I would like to know why this DFT functional has been used throughout in this study. Also, I would like to know why the 6-311+G(d) basis set was used. The authors have done some benchmarking calculations or not?

5.       In this work, the authors have compared the calculated spectrum with that obtained from “Gaussian vibronic treatment” (Figure 5). Please describe briefly the differences between the present vibronic model method and “Gaussian vibronic treatment” so as that the readers can understand the reason for the observed spectral difference. Also, the present vibronic model still underestimates the absorption strength as can be seen from Figure 5. Please add detailed descriptions for possible reasons. In particular, I would like to know if the spectrum calculated by "Gaussian vibronic treatment” includes the correction of the 0-0 transition.
